# Morphology of Meibomian Glands in a 65-Year-Old Norwegian Population without Dry Eye Disease

**DOI:** 10.3390/jcm11030527

**Published:** 2022-01-20

**Authors:** Xiangjun Chen, Reza A. Badian, Håvard Hynne, Behzod Tashbayev, Lene Hystad Hove, Janicke Liaaen Jensen, Tor Paaske Utheim

**Affiliations:** 1Department of Oral Surgery and Oral Medicine, Faculty of Dentistry, University of Oslo, Geitmyrsveien 71, 0455 Oslo, Norway; havard.hynne@odont.uio.no (H.H.); bektashbayev@gmail.com (B.T.); j.c.l.jensen@odont.uio.no (J.L.J.); 2Department of Ophthalmology, Sørlandet Hospital Arendal, 4838 Arendal, Norway; 3Department of Medical Biochemistry, Oslo University Hospital, 0450 Oslo, Norway; rezabadian@gmail.com (R.A.B.); utheim2@gmail.com (T.P.U.); 4The Norwegian Dry Eye Clinic, 0366 Oslo, Norway; 5Department of Cariology and Gerodontology, Faculty of Dentistry, University of Oslo, 0315 Oslo, Norway; l.h.hove@odont.uio.no; 6Department of Oral Biology, Faculty of Dentistry, University of Oslo, 0315 Oslo, Norway; 7Department of Ophthalmology, Oslo University Hospital, 0450 Oslo, Norway

**Keywords:** meibomian gland, meibomian gland dysfunction, dry eye disease, meibography

## Abstract

Analyses of meibography may help in the diagnosis, prevention, and management of meibomian gland dysfunction (MGD). However, there is currently a paucity of data regarding meibography analyses in the young elderly populations in the Nordic countries. In the current study, meibography of the upper and lower eyelids of 117 65-year-old residents in Oslo, Norway, who did not fulfil the diagnosis of dry eye disease (DED) were analysed. Meibomian gland (MG) dropout and tarsal areas were measured semi-automatically using ImageJ software. The relationship between morphological features of the MGs and clinical dry eye tests was examined. The median percent MG dropout was 26.1% and 40.7% in the upper and lower eyelids, respectively. There was no significant difference between males and females. None of the MG morphological parameters demonstrated significant values in discriminating abnormal dry eye symptom loads or MGD diagnosis from the normal loads. We therefore concluded that moderate MG atrophy was common among the Norwegian population of 65-year-olds without DED and showed no sexual differences. Meibography alone cannot discriminate MGD from non-MGD; thus, both morphological and functional MG tests are necessary when screening for MGD.

## 1. Introduction

Meibomian glands (MGs) are modified sebaceous glands vertically embedded in the tarsal plates of the upper and lower eyelids. It is reported that 25–40 MGs are present in the upper tarsus and 20–30 in the lower tarsus [1]. Adequate structure and function of MGs is critical for a healthy ocular surface because the lipids secreted by the MGs, the meibum, are essential components of the tear film. The lipids retard tears from excessive evaporation and function as lubricants for the eyelids during blinking. Therefore, the absence or altered integrity of the meibum may result in decreased tear stability and increased tear evaporation, as well as a loss of lubrication for the ocular surface. Meibomian gland dysfunction (MGD) is defined as obstruction of the terminal duct of the MGs and/or quantitative/qualitative changes in glandular secretion [2] and is the most common cause of evaporative dry eye disease (DED) [3,4]. The reported prevalence of MGD is 3.5–19.9% in Caucasians and was found to be higher than 60% in some Asian population-based studies [4].

It is recommended that a combination of both functional and morphological assessments of the MG is used to evaluate MGD [5]. The functionality of the MGs can be examined by the expressibility and quality of the meibum. Alterations of the MG morphology are clinically less apparent and conceivably underreported. Noncontact infrared meibography enables the visualization of structural appearance and atrophy/dropout of the MGs through the use of an infrared charge-coupled video camera and infrared transmitting filter. Several studies have indicated that alterations in MG morphology can be associated with clinical dry eye tests, suggesting that morphometric analysis of the MG using meibography may be useful for assessing ocular surface conditions [6,7,8].

Ageing is believed to be one of the most influential risk factors of MG atrophy [6,8,9,10,11]. A study by Nien et al. showed decreased meibocyte differentiation and cell cycling in older compared with younger subjects. Altered peroxisome proliferator-activated receptor γ (PPARγ) signaling may lead to acinar atrophy and the development of an age-related hyposecretory MGD [10]. An increased prevalence of atrophy with age may make the assessment of MGs of added importance in senior populations.

The pathogenesis of MGD is still poorly understood. Analyses of meibography may help in the diagnosis, prevention, and management of MGD. However, there is currently a paucity of data regarding meibography analyses in the young elderly populations in the Nordic countries. The lack of normative data may create a gap in the early diagnosis of MGD. Our primary purpose was to describe meibography characteristics in a Norwegian cohort of 65-year-olds without a DED diagnosis by performing a cross-sectional study with 117 Oslo residents to establish baseline values in this well-defined population. A secondary aim was to investigate the relationship between MG features and other dry eye-related parameters.

## 2. Methods

This cross-sectional study is a sub-study on ocular health as an extension to a larger project focusing on oral health in 65-year-olds in Oslo, Norway (OM65) [12]. A total of 457 Oslo residents attended the examination in the OM65 study, and they were all invited to participate in the current study to examine the health status of their ocular surfaces. The study was approved by the Norwegian Regional Committee for Medical and Health Research Ethics (REK 2018/1383). It was conducted at the Norwegian Dry Eye Clinic and was performed in compliance with the tenets of the Declaration of Helsinki. Written informed consent was obtained from all subjects before examination. By March 2020, when the recruitment was stopped due to the COVID-19 pandemic, 150 participants had been enrolled.

All participants were examined by two experienced ophthalmologists in the field of ocular surface disease. The examination for the current study included an Ocular Surface Disease Index (OSDI) dry eye questionnaire, tear film osmolarity measured with the I-PEN ^®^ Tear Osmolarity System (I-MED Pharma Inc., Dollard-des-Ormeaux, QC, Canada), fluorescein tear break-up time (TFBUT) after instillation of 5 µL 2% fluorescein, ocular surface fluorescein staining (OSS) of the cornea and the nasal and temporal conjunctiva graded according to the Oxford system (score ranges 0–5 for each of the three regions, and 0–15 for all three regions), Schirmer test without anaesthesia, slit-lamp observation of lid margin abnormalities, meibomian gland functionality and non-contact infrared meibography. Lid margin abnormality (LMA) was scored from 0 to 4 based on the number of the following four abnormalities: irregularity of the lid margin, lid margin telangiectasia, lid margin hyperaemia and thickening of the eyelid margin. Meibomian gland functionality was evaluated under slit lamp by pressing the central five MGs in the lower lids of each eye. Meibomian gland expressibility (ME) was recorded as the number of glands that secrete meibum (score range: 0–5). The quality of meibum (MQ) secreted by each gland was graded according to a four-point score (0: clear; 1: cloudy; 2: granular; 3: toothpaste), and the average score of the five glands was calculated. Meibography was performed using a Keratograph 5M machine (Oculus GmbH, Wetzlar, Germany).

Further analyses of meibography images were performed by one well-trained observer who was masked to the clinical data. The current study adopted a meibography grading protocol similar to the one used by Daniel et al. [13]. Briefly, the areas of the total and middle-third of the tarsal plate, as well as the dropout areas of the total and middle-third of the tarsal plate, were measured using ImageJ software (Figure 1). The dropout rate in the entire tarsal area and the middle-third area was calculated as (Area_Dropout_/Area_Tarsal plate_ × 100%). Fluffy areas (areas where individual MGs displayed a lack of normal architecture, with only an amorphous white substance) and ghost glands (MGs appeared pale and blurry without normal architecture) were counted as areas of dropout. The meibomian glands on the side edges of the lids can be prone to inaccuracies to the observer; therefore, further assessment of specific features of the MGs, such as the number of distorted (glands not following the parallel course of normal glands but with torsion ≤45°), tortuous (glands with torsion >45°), hooked (glands curling back at the distal end), dropout, shortened, overlapping and ghost, was performed in the middle-third area of each eyelid. The presence of features such as tadpoling (glands thicker at the eyelid margin but tapering and thinning out distally), abnormal gap (gap between two adjoining glands at least twice that of a normal gland), fluffy area and no extension to the lid margin were checked over the whole tarsal area. Representative examples of these features are shown in Figure 2.

Statistical analyses were performed using the commercial software SPSS, version 27 (IBM, Chicago, IL, USA). The average values of the two eyes were used for analyses. The normality of data distribution was assessed using a Shapiro–Wilk test. Quantitative variables were described using the median and interquartile range, whereas categorical variables were reported as frequencies. The Mann–Whitney U test and Student’s *t*-test were used for comparison of non-normal and normally distributed variables between males and females, respectively. A Wilcoxon signed-rank test and paired sample *t*-test were used for comparison between upper and lower lids. A Chi-squared test was applied to analyze the percentage differences, and Spearman rank correlation coefficients (Rs) were calculated to measure the strength of correlations between parameters. The ability of meibomian gland morphological measurements to predict abnormal dry eye symptom loads, as well as diagnosis of MGD, was analyzed by a receiver operating characteristic (ROC) curve. Abnormal dry eye symptom load was defined as an OSDI score ≥ 13. Diagnosis of DED was made when OSDI ≥ 13, accompanied by TFBUT < 10 s and/or OSS ≥ 1, as suggested by DEWS II [14]. The presence of MGD was defined as (1) at least one lid margin abnormality (LMA ≥ 1), and (2) meibomian gland functional changes as revealed by the detection of reduced meibum expression (ME < 5) and/or meibum quality (MQ ≥ 1). A general linear model was used to analyze the risk factors for MG dropout, adjusted for sex. A *p*-value of <0.05 was considered statistically significant.

## 3. Results

Meibography was performed in 147 subjects, among whom 30 were diagnosed as DED, based on the aforementioned criteria and were, thus, excluded from the current study. Hence, data from 117 non-DED subjects (60 men and 57 women) were included for further analyses. Among the 468 available meibography images, 393 were of sufficient quality for interpretation. These images included 98 upper lids and 99 lower lids of the right eye and 89 upper lids and 107 lower lids of the left eye. The reasons for exclusion were as follows: insufficient lid eversion, low image quality and/or fingers obscuring the view of the eyelids.

### 3.1. Study Subjects’ Clinical Characteristics

The demographic and clinical characteristics of the participants are summarized in Table 1 and Table 2. The ME was worse in females than in males (median, 3.5 vs. 4, *p* = 0.040). No statistically significant differences were found in other clinical dry eye tests.

Based on the aforementioned criteria, 10.3% (12 out of 117) had abnormal OSDI, and 69.0% (80 out of 116) had MGD.

### 3.2. Meibomian Gland Morphological Features

The morphological features of the MGs are listed in Table 3 and Table 4 and Figure 3. The majority of the study population had dropout in the range of 26–50% in both upper and lower eyelids, whereas none of them had a dropout of >75%. The level of dropout was higher in the lower lids than the upper ones (median, 40.7% vs. 26.1%, *p* < 0.001). Similarly, in the middle-third of the lids, the atrophy was 24.9% in the upper lids and 43.5% in the lower lids (*p* < 0.001). There were no statistically significant differences between male and female subjects. In both males and females, upper lids had more visible MG features than lower lids, except that a lack of extension of MGs to the lid margin and fluffy areas were observed in a higher proportion of lower lids (Table 4).

### 3.3. Correlations between Meibomian Gland Morphology and Other Dry Eye Tests

Most of the morphological features of the MGs were not significantly associated with other clinical dry eye tests (Appendix A). A higher percent of dropout in the upper lids was associated with more OSS (r = 0.246, *p* = 0.012), whereas more dropout in the lower lids correlated with lower Schirmer test values (r = −0.206, *p* = 0.029) and a lower osmolarity level (r = −0.353, *p* = 0.011).

No significant differences in MG dropout were observed between subjects with and without abnormal symptom load or between subjects with and without MGD (Table 5). Further, we attempted to determine whether the morphological characteristics could discriminate abnormal dry eye symptom loads and the diagnosis of MGD by employing ROC curve analysis. None of the quantitative variables demonstrated significant values in discriminating abnormal dry eye symptom loads or MGD diagnosis (*p* > 0.05).

### 3.4. Meibomian Gland Dropout Risk Factors

We further analyzed the effect of race, systemic disease, concomitant use of oral and ocular medication, smoking status (current/past smoker vs non-smoker) and contact lens use on meibomian gland dropout, adjusted for sex. General linear model analyses showed that past or current smokers had a slightly higher MG dropout rate in the upper lids (median 27.4%, Q1–Q3: 23.2–41.4% vs median 25.8%, Q1–Q3: 18.3–29.0%, *p* = 0.003) than those who never smoked. No significant association was observed between MG dropout and other factors.

## 4. Discussion

Evaluation of MG morphology may be helpful in assessing MGD and evaluating the effect of therapeutic interventions, such as eyelid hygiene treatment and intense pulsed-light therapy [15,16]. It is important to standardize normal MG morphological values in age-specified populations because such an approach will help to recognize subjects at risk of developing DED. Using non-contact infrared meibography, our study demonstrated that substantial MG atrophy is present in the young elderly Norwegian population without DED. No sex-related differences were detected in the current study.

In obstructive MGD, hyperkeratinization of the ductal epithelium and increased meibum viscosity can lead to intraglandular cystic dilatation, meibocyte atrophy and gland dropout [4]. However, MG dropout can also occur as an age-related atrophic process [17]. Accordingly, our data showed significant MG atrophy in subjects with and without MGD. One intriguing observation was the substantial variety in MG dropout morphology in our study population, even in subjects without significant dry eye symptoms (upper lids range: 11–51%; lower lids range: 20–64%) and in the absence of MGD (upper lids range: 13–47%; lower lids range: 20–61%). This variety may warrant additional counselling on the potential future risk of developing MGD and DED in these subjects.

Although a study by Pult et al. showed that the MG dropout in the upper and lower lids was correlated [18], the difference between lids emphasizes the importance of evaluating both lids. Similar to previous reports [4,13], our study showed a higher number of MGs in the upper than lower lids. Accordingly, the upper lids had more visible MG alterations than the lower lids, such as the number of distorted, tortuous, and hooked MGs. On the other hand, no extension of MGs to the lid margin and fluffy areas were more common in the lower lids. These findings agree with Daniel and colleagues’ results [13]. The MG dropout rate was higher in the lower compared to upper lids, which is inconsistent with previous research in subjects with or without DED or MGD [13,18,19].

Meibomian glands are hormone target organs. Androgen has been demonstrated to play critical roles in lipid biosynthesis, epithelial keratinization, and interaction with other hormones to regulate their functions in the MG [20,21,22]. Thus, androgen deficiency may promote MGD [23,24]. Oestrogen, on the other hand, opposes some actions of androgen in the MGs [24,25]. Therefore, there is a higher risk of developing DED among women. However, sex differences in DED may lessen with more advanced age [24]. In the present study, no significant difference in MG morphology between males and females was detected, potentially because the most significant decline in androgen production occurs before the age of 60 years [26], and as Yeotikar et al. suggested [27], both men and women at 65 years of age are androgen deficient, resulting in similar alterations in MGs. Our results of no sex differences were in accordance with several previous publications [9,27,28,29,30]. However, they are not in line with other studies [8,11,31]. For example, Arita et al. showed that the degree of meibomian gland atrophy tended to be more severe in elderly men than in women (60- to 69-year-olds, and those over 80 years) [8]. Their conclusion was supported by a recent analysis of MG atrophy in middle-aged to elderly patients with cataracts [11]. Moreover, Den et al. reported that men above the age of 70 more likely show changes in their MGs (dropout of the gland structure in more than one half of the lower lid) than women [31], whereas no sex-related differences were found in subjects below 70 years. The reasons causing the inconsistencies among studies may include differences in inclusion/exclusion criteria for the study population; eyelids assessed (i.e., lower lids, upper lids, or both); or the methods used in the evaluation of MG dropout (i.e., different subjective meiboscore scales used vs semiautomatic analysis).

Whether correlations exist between changes in MGs and tear film-related parameters has been a controversial subject. Some studies showed no significant association between MG dropout and OSDI, Schirmer test, OSS or TFBUT [8,19,27,31,32], whereas others revealed correlations between MG dropout and lid margin abnormality score [8], OSDI [9,18,28,33], Schirmer test and TFBUT [34] and MQ [19,35]. In the current study, the majority of MG morphological parameters did not show significant correlations with clinical dry eye tests, except that a higher dropout rate in the upper lids was associated with higher OSS, whereas increased dropout in the lower lids correlated with lower Schirmer test values and decreased osmolarity levels. The disparities among studies may be partly attributed to the limited measurement reliability in meibography analysis as well as in clinical dry eye tests. Although the semi-objective analysis of MG dropout was reported to be more reliable and more sensitive than the subjective meiboscore [33,36], the quality of images and the observer’s subjective quantifications could lead to a limited repeatability of the measurements [13]. Moreover, the fact that the current study included only subjects without DED might have contributed to the lack of correlation between the changes in MGs and tear film-related parameters.

Some previous studies showed that MG dropout was higher in MGD patients compared to non-MGD ones and that MG dropout could be used to discriminate MGD from non-MGD or DED from non-DED [5,28,33]. However, in our study population, the MG dropout in subjects with MGD did not differ from those without. The precise reason causing this inconsistency is not clear. It may be partly attributed to differences in the study population and in the MGD diagnostic criteria used among studies. In the current study, only 65-year-old subjects who did not meet the diagnosis of DED were included. With the relatively small differences in ME and MQ between MGD and non-MGD patients (3.5 ± 0.9 vs. 4.3 ± 0.9, and 1.5 ± 0.7 vs. 0.5 ± 0.7, respectively), the age-related MG morphological changes might have outweighed the differences related to MGD.

Similar to Machalinska et al. [9], we showed a slightly higher MG loss in subjects with a history of smoking than those who never smoked. Remarkably, no association was observed with systemic disease or concomitant use of oral or ocular medications, contrasting with some other studies that identified diabetes [37,38], hormone replacement therapy [9] and use of antiallergy drugs [9] as risk factors for MG loss. The relatively small sample size of the subjects might have caused this inconsistency.

There are some limitations of the present study. This research may have a potential selection bias because the study subjects were recruited from the participants in the OM65 project. That all participants were at the same age of 65 years can be seen both as an advantage because age would then not be a confounding factor when comparing females and males, and a disadvantage as a wider age range may be more representative for the young elderly population.

In conclusion, this study showed that in the Norwegian population of 65-year-olds without DED, moderate MG atrophy was common and without sexual differences. The morphological features presented in the current study may serve as normative values for young elderly Norwegians. Moreover, our data show that most MG morphological features were not associated with other dry eye tests and that meibography alone cannot discriminate MGD from non-MGD; thus, both morphological and functional tests of MGs are necessary when screening patients at risk of MGD.

## Figures and Tables

**Figure 1 jcm-11-00527-f001:**
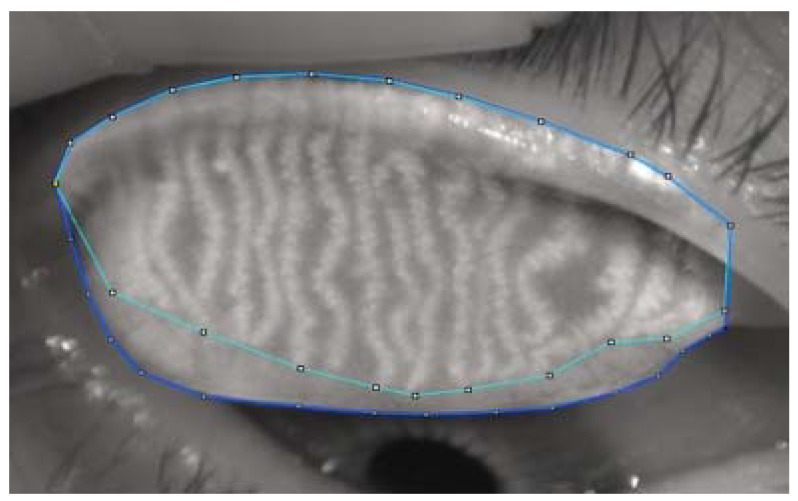
Image J-assisted Meibomian gland atrophy analysis. The upper and lower outlined regions represent gland and dropout areas, respectively.

**Figure 2 jcm-11-00527-f002:**
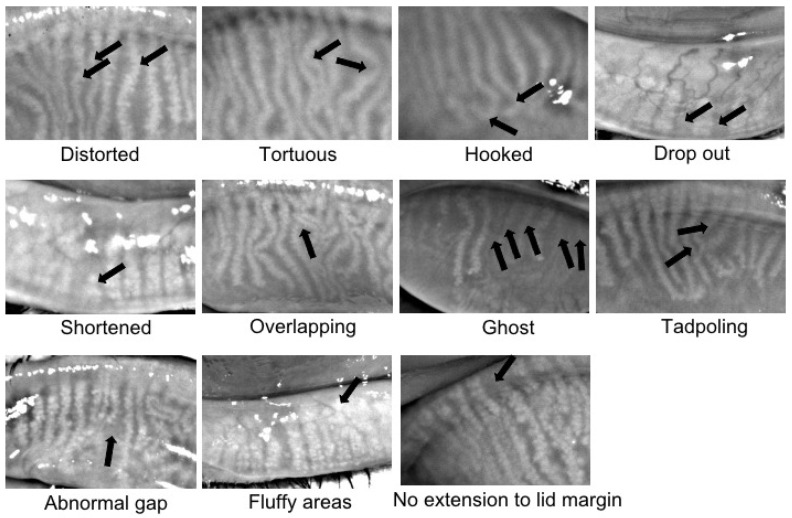
Representative images of meibomian gland morphological changes.

**Figure 3 jcm-11-00527-f003:**
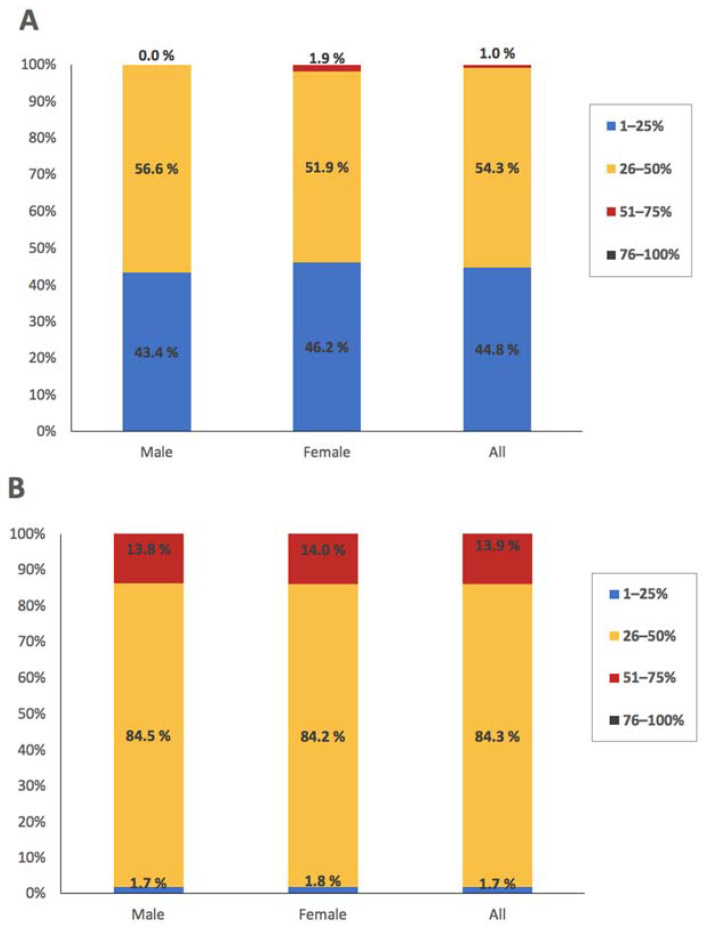
Distribution of meibomian gland dropout (**A**) in the upper lids and (**B**) in the lower lids.

**Table 1 jcm-11-00527-t001:** Demographic characteristics of study subjects.

	All		Male		Female		*p*-Value *
	*n*	%	*n*	%	*n*	%	
Ethnicity	117		60		57		0.062
West European	109	93.2	53	88.3	56	98.2	
Other	8	6.8	7	11.7	1	1.8	
Systemic disease	117		60		57		
Diabetes	7	6.0	5	8.3	2	3.5	0.440
Hypertension	28	23.9	16	26.7	12	21.1	0.521
Rheumatic disease	11	9.4	4	6.7	7	12.3	0.354
Concomitant use of oral/ocular medication	117		60		57		
Beta-blockers	11	9.4	4	6.7	7	12.3	0.354
Diuretics	6	5.1	2	3.3	4	7.0	0.431
Antidepressants	3	2.6	1	1.7	2	3.5	0.612
Lipid lowering	30	25.6	19	31.7	11	25.6	0.143
Antihistamines	15	12.8	7	11.7	8	14.0	0.786
Postmenopausal hormonal	8	6.8	0	0	8	14.0	0.002
Use of contact lenses	6	5.1	3	5.0	3	5.3	1.000
Use of topical ocular lubricants	9	7.7	2	3.3	7	12.3	0.089
Smoking habits	117		60		57		0.854
Never	57	48.7	30	50.0	27	47.4	
Past or current smoking	60	51.3	30	50.0	30	52.6	

* *p*-Value from Chi-square test.

**Table 2 jcm-11-00527-t002:** Results of clinical dry eye tests.

	All Median (Q1, Q3)	Male Median (Q1, Q3)	Female Median (Q1, Q3)	*p*-Value *
OSDI (*n* = 117)	2.1 (0, 6.5)	1.0 (0, 4.6)	2.3 (0, 7.6)	0.203
Osmolarity (*n* = 51)	319.0 (307.0, 321.0)	320.0 (307.3, 324.8)	316.8 (306.8, 328.5)	0.653
TFBUT (*n* = 113)	7.5 (4.3, 13.0)	8.5 (4.0, 13.3)	6.5 (4.4, 12.1)	0.774
OSS (*n* = 117)	0.5 (0, 1.0)	0.3 (0, 1.0)	0.5 (0, 1.0)	0.640
Schirmer (*n* = 115)	11.3 (7.5, 19.0)	9.8 (6.5, 14.6)	11.5 (8.0, 22.5)	0.194
ME (*n* = 116)	4.0 (3.0, 5.0)	4.0 (3.5, 5.0)	3.5 (3.0, 4.5)	0.040
MQ (*n* = 116)	1.0 (0.4, 2.0)	1.0 (0.3, 2.0)	1.0 (0.5, 2.0)	0.742
LMA (*n* = 117)	3.0 (1.0, 4.0)	3.0 (1.0, 4.0)	3.0 (1.0, 4.0)	0.747

OSDI = Ocular Surface Disease Index; TFBUT = tear film break-up time; OSS = ocular surface staining; ME = meibum expressibility; MQ = meibum quality; LMA = lid margin abnormality score. * *p*-value from Mann–Whitney test.

**Table 3 jcm-11-00527-t003:** Percent dropout of meibomian gland.

Dropout	AllMedian (Q1, Q3)	MaleMedian (Q1, Q3)	FemaleMedian (Q1, Q3)	*p*-Value *
Upper lids (*n* = 132)	26.1% (21.6%, 33.1%)	26.1% (20.9%, 33.2%)	26.1% (22.0%, 33.1%)	0.946
Lower lids (*n* = 142)	40.7% (35.9%, 47.1%)	40.7% (37.2%, 44.6%)	39.5% (34.9%, 48.1%)	0.612
*p*-value ^#^	<0.001	<0.001	<0.001	
Middle upper lids (*n* = 132)	24.9% (18.6%, 35.4%)	26.0% (19.4%, 38.8%)	24.0% (17.5%, 34.3%)	0.228
Middle lower lids (*n* = 142)	43.5% (38.2%, 48.8%)	43.8% (39.3%, 47.2%)	41.6% (35.8%, 50.5%)	0.465
*p*-value ^#^	<0.001	<0.001	<0.001	

* Differences between males and females evaluated with Student *t*-test with normally distributed data and with Mann–Whitney test with non-normally distributed data. ^#^ Differences between upper and lower lids evaluated with Wilcoxon signed rank test.

**Table 4 jcm-11-00527-t004:** Frequency of meibomian gland features in the middle-third of the lids.

Features	All			Male			Female			*p*-Value	
	UL	LL	*p **	UL	LL	*p **	UL	LL	*p **	UL ^1^	LL ^1^
	Median (Q1, Q3)	Median (Q1, Q3)		Median (Q1, Q3)	Median (Q1, Q3)		Median (Q1, Q3)	Median (Q1, Q3)			
Total number	8.0 (7.5, 8.5)	6.5 (6, 7.0)	<0.001	8.0 (7.3, 8.5)	6.5 (6.0, 7.0)		8.0 (7.5, 8.5)	6.5 (6.0, 7.0)		0.292	0.305
Distorted	3.0 (2.0, 3.5)	2.0 (1.5, 3.0)	<0.001	3.0 (2.0, 3.5)	2.0 (1.5, 2.6)		2.8 (2.0, 3.4)	2.5 (1.5, 3.0)		0.327	0.474
Tortuous	3.0 (2.5, 4.0)	0 (0, 0.5)	<0.001	3.0 (2.5, 3.8)	0 (0, 0.5)		3.0 (2.5, 4.0)	0 (0, 1.0)		0.797	0.739
Hooked	1.0 (0.5, 2.0)	0 (0, 0)	<0.001	1.0 (0.5, 2.0)	0 (0, 0)		1.0 (0.5, 2.0)	0 (0, 0.5)		0.762	0.014
Short	3.0 (2.5, 4.5)	2.0 (2.0, 3.0)	<0.001	3.0 (2.5, 4.5)	2.0 (1.9, 3.0)		3.0 (2.5, 4.5)	2.0 (2.0, 3.0)		0.596	0.826
Overlap	3.0 (2.0, 4.5)	0 (0, 1.5)	<0.001	3.0 (2.0, 4.5)	0 (0, 1.1)		2.5 (1.1, 4.4)	0 (0, 1.8)		0.164	0.961
Ghost	0 (0, 2.5)	0 (0, 0)	<0.001	0 (0, 2.5)	0 (0, 0)		0 (0, 1.4)	0 (0, 0)		0.367	0.801
Drop-out	0 (0, 0)	0 (0, 0.5)	0.019	0 (0, 0)	0 (0, 0.5)		0 (0, 0)	0 (0.0, 0.5)		0.532	0.687
**Other Features**	***n* (%)**	***n* (%)**	** *p* ^#^ **	***n* (%)**	***n* (%)**	** *p* ^#^ **	***n* (%)**	***n* (%)**	** *p* ^#^ **	***p*-Value** **UL ^2^**	***p*-Value** **LL ^2^**
Tadpoling (yes)	20 (18.9%)	7 (6.1%)	0.0473	10 (18.5%)	4 (6.9%)	1.000	10 (19.2%)	3 (5.3%)	1.000	1.000	1.000
Gaps (yes)	101 (96.2)	91 (64.1%)	0.015	52 (98.1%)	38 (65.5%)	0.377	49 (94.2%)	39 (68.4%)	0.037	0.363	0.843
No lid margin extent (yes)	5 (4.8%)	36 (31.3%)	0.002	2 (3.8%)	18 (31.0%)	0.076	3 (5.8%)	18 (31.6%)	0.031	0.678	1.000
Fluffy areas (yes)	71 (67.6%)	114 (99.1%)	0.324	33 (62.3%)	57 (98.3%)	0.377	38 (73.1%)	57 (100%)	NA	0.298	1.000

UL = upper lid; LL = lower lid. ^1.^ Differences between males and females evaluated with Student *t*-test with normally distributed data and with Mann–Whitney test with non-normally distributed data. ^2.^ Differences between males and females evaluated with Chi-square test. * Differences between upper and lower lids evaluated with Wilcoxon signed rank test. ^#^ Differences between upper and lower lids evaluated with Chi-square test.

**Table 5 jcm-11-00527-t005:** Meibomian gland atrophy in different groups of subjects.

	Percent Dropout	
	Upper Lids	Lower Lids
Symptomatic	24.6% (17.8%, 38.0%)	39.9% (35.9%, 43.8%)
Asymptomatic	26.2% (21.7%, 32.9%)	40.7% (36.1%, 47.8%)
*p*-value ^1^	0.823	0.628
MGD	26.3% (22.7%, 33.1%)	40.7% (34.9%, 47.8%)
Non-MGD	25.3% (19.0%, 35.3%)	40.1% (37.0%, 44.0%)
*p*-value ^1^	0.693	0.965

Data are presented as median (Q1, Q3). MGD = meibomian gland dysfunction. Symptomatic group included subjects with OSDI score ≥ 13, while asymptomatic group included subjects with OSDI score < 13. ^1.^ Differences between males and females evaluated with Student’s *t*-test with normally distributed data and with Mann–Whitney test with non-normally distributed data.

## Data Availability

The data presented in this study are available on request from the corresponding author.

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
