# Peer review of "Morphology of Meibomian Glands in a 65-Year-Old Norwegian Population without Dry Eye Disease"

_jcm, 2022, doi:10.3390/jcm11030527_

Round 1

Reviewer 1 Report

This manuscript investigated the morphology of meibomian gland in 117 subjects of 65-year old Norweigian population without dry eye disease. The normotive data is important because it can serve as a standard. Therefore, the meibomian gland morphological analysis of normal 65-year-old subjects without dry eye disease conducted in this study can be meaningful. I would like to point out a few points to develop the current manuscript.

1) Page 4, line 131, The authors defined dry eye disease according to DEWS II. However, although the authors measured tear osmolarity, they were excluded osmolarity from the diagnostic criteria. It is required to revise the diagnostic criteria according to DEWS II.

2) Page 2, line 84. “ocular surface fluorescein staining (OSS) of the cornea and conjunctiva graded according to the Oxford system (range 0–15)”

The oxford system is divided into 0 to 5 grades. This part needs to be corrected. Is the NEI score used?

3) In Table 3 & 4, Add the p-value for the comparison between the upper and lower lids.

4) In table 5, Does symptomatic mean exceeding OSDI 13? Add an explanation to the table legend.

5) There was no significant difference in dropout percent between MGD and non-MGD subjects. It seems that further explanation is necessary.

Author Response

We thank the reviewer for stating the study is meaningful. The reviewer also has a set of comments:

1) Page 4, line 131, The authors defined dry eye disease according to DEWS II. However, although the authors measured tear osmolarity, they were excluded osmolarity from the diagnostic criteria. It is required to revise the diagnostic criteria according to DEWS II.

Response: We agree with the reviewer that DEWS II did propose tear hyperosmolarity as a disease marker and suggest inclusion of osmolarity measurements as one of the diagnostic tools. However, due to delivery supplies issues from the company in Canada for the I-PEN Tear Osmolarity System, the osmolarity measurement was performed in only 66 out of the 150 examined subjects. This made it difficult to include osmolarity in the diagnostic criteria for dry eye disease in the current study. Moreover, while some studies demonstrated elevated tear osmolarity in patients with dry eye disease, compared to healthy controls3, 5, others found the osmolarity measurements performed by the currently commercially available devices to be variable and advised that they could not be used as a key indicator for DED.2, 4. Therefore, the current study we chose to present the measured osmolarity data but not include them as part of the dry eye disease diagnostic criteria.

2) Page 2, line 84. “ocular surface fluorescein staining (OSS) of the cornea and conjunctiva graded according to the Oxford system (range 0–15)”

The oxford system is divided into 0 to 5 grades. This part needs to be corrected. Is the NEI score used?

Response: The Oxford system scaled staining in the regions of the cornea, and nasal and temporal conjunctiva, with a score of 0-5 representing different severity of the staining in each region. Therefore, the total score for ocular surface staining amounted to 0-15 https://www.tearfilm.org/dewsreport/pdfs/Staining%20grading%20Oxford%20Schema%20(Bron).pdf. We have now modified the sentence to clarify this issue: “ocular surface fluorescein staining (OSS) of the cornea and the nasal and temporal conjunctiva graded according to the Oxford system (score ranges 0-5 for each of the three regions, and 0–15 for all three regions)”

3) In Table 3 & 4, Add the p-value for the comparison between the upper and lower lids.

Response: We thank the reviewer for the suggestion, and the p-values have now been added to Tables 3 & 4.

4) In table 5, Does symptomatic mean exceeding OSDI 13? Add an explanation to the table legend.

Response: We thank reviewer for the suggestion. The sentence “Symptomatic group included subjects with OSDI score≥ 13, while asymptomatic group included subjects with OSDI< 13.” has been added to the table legend.

5) There was no significant difference in dropout percent between MGD and non-MGD subjects. It seems that further explanation is necessary.

Response: This might be related to the study population (65-year-olds that did not meet DED diagnosis in the current study). We have added the following sentences to the discussion: In the current study, only 65-year-old subjects who did not meet the diagnosis of DED were included. With the relatively small differences in ME and MQ between MGD and non-MGD patients (3.5±0.9 vs. 4.3±0.9, and 1.5±0.7 vs. 0.5±0.7, respectively), the age-related MG morphological changes might have overweighted the differences related to MGD.

Reviewer 2 Report

Well-made manuscript, but two questions are required to be answered and minor editing is needed (see the attached file).

1. Why did you exclude the dry eye disease patients in this study?

2. I think the reason why the morphology of the MGs were not significantly related to clinical dry eye tests in the manuscript is possibly due to the selection of subjects without dry eye. What do think about that?

Author Response

We thank the reviewer for the positive comments. The reviewer also has a set of comments.

  1. Why did you exclude the dry eye disease patients in this study?

Response: The current study aimed to establish baseline values in meibomian gland morphology among young elderly subjects with relatively healthy ocular surface, that future studies investigating ocular surface disease may use as reference. MGD is highly prevalent among patients with dry eye disease,1 and MGD-related meibomian gland morphological changes may influence the baseline values. Therefore, patients with dry eye disease were excluded from the current study.

  1. I think the reason why the morphology of the MGs were not significantly related to clinical dry eye tests in the manuscript is possibly due to the selection of subjects without dry eye. What do think about that?

Response: We agree with the reviewer. One sentence “Moreover, the fact that the current study included only subjects without DED might have contributed to the lack of correlation between the changes of MGs and tear film-related parameters.” has now been added to the Discussion part.

3) The word "unclear" has been replaced by "clear" in the Discussion part.